# Nanoscale Surface Refinement of CoCrMo Alloy for Artificial Knee Joints via Chemical Mechanical Polishing

**DOI:** 10.3390/ma17010008

**Published:** 2023-12-19

**Authors:** Hanji Zhang, Jiangliang Zhang, Jinghui Lai, Yilin Chen, Mengqiang Tian, Guofeng Pan, Xueli Yang, Yuhang Qi

**Affiliations:** 1Department of Orthopedics, Tianjin Union Medical Center, Nankai University Affiliated Hospital, Tianjin 300121, China; zhanghanji@umc.net.cn; 2School of Electronics and Information Engineering, Hebei University of Technology, Tianjin 300130, China; 15185054254@163.com (J.Z.); l1516570@163.com (J.L.); pgf@hebut.edu.cn (G.P.); 3The Second Clinical College, China Medical University, Shenyang 110001, China; 2019341120@stu.cmu.edu.cn

**Keywords:** chemical mechanical polishing, CoCrMo alloy, surface roughness, artificial knee joint, friction behavior

## Abstract

In this study, we address the challenge of surface roughness in CoCrMo alloys, typically used in artificial knee joints, which can initiate a cascade of biological responses causing inflammation, osteolysis, joint instability, and increased susceptibility to infection. We propose the application of a chemical mechanical polishing (CMP) technique, using an ecologically responsible slurry composed of 4 wt% SiO_2_, 0.3 wt% H_2_O_2_, 1.0 wt% glycine, and 0.05 wt% benzotriazole. Our innovative approach demonstrated significant improvements, achieving a material removal rate of 30.9 nm/min and reducing the arithmetic mean roughness from 20.76 nm to 0.25 nm, thereby enhancing the nanoscale surface quality of the artificial knee joint alloy. The smoother surface is attributed to a decrease in corrosion potential to 0.18 V and a reduction in corrosion current density from 9.55 µA/cm^2^ to 4.49 µA/cm^2^ with the addition of BTA, evidenced by electrochemical tests. Furthermore, the preservation of the phase structure of the CoCrMo alloy, as confirmed by XRD analysis and elemental mapping, ensures the structural integrity of the treated surfaces. These outcomes and our simulation results demonstrate the effectiveness of our CMP method in engineering surface treatments for artificial knee joints to optimize friction behavior and potentially extend their lifespans.

## 1. Introduction

Artificial knee joint failure can be attributed to numerous causes, among which severe joint wear stands out as a major contributor [1,2,3]. This mainly consists of adhesive wear and abrasive wear, and it typically generates a large number of wear particles. The prolonged presence of these particles within the body can incite necrosis of the cells surrounding the joint, potentially leading to osteolysis [4]. Moreover, the interactions of these wear particles within the body can also negatively impact the internal organs, kidneys, and peripheral tissues, thereby increasing the health risks for patients [5,6,7]. Ultimately, these wear particles serve as a fundamental cause of aseptic loosening post-hip arthroplasty, necessitating additional surgeries and revisions in severe cases, which in turn augment the patient’s economic burden and psychological distress.

Consequently, the fabrication of artificial joint prostheses with higher surface smoothness [8], particularly those composed of CoCrMo alloys, to minimize the generation of particles due to friction wear [9], is anticipated to significantly improve the lifespan and performance of joint prostheses. Current methods of polishing artificial joints primarily include mechanical polishing [10], electrolytic polishing [11], magnetorheological polishing [12,13], and ultrasonic polishing [14,15]. However, mechanical polishing may introduce microscopic scratches, increasing surface roughness. Electrolytic polishing requires complex power sources and precise parameter control, making control of the process challenging. Improper handling could result in excessive polishing or even burn the surface. Magnetorheological polishing employs a controllable magnetic field to achieve precise polishing. Despite its potential for high-precision polishing, its main downside lies in the high cost of the magnetorheological fluid and the possible degradation of the fluid during use. Moreover, this method is only applicable to magnetic materials, limiting its scope of application. Ultrasonic polishing utilizes ultrasonic vibrations to eliminate surface defects in materials. However, this method has low efficiency when dealing with parts that have large surface areas.

Inspired by the polishing process for integrated circuits, polishing processes with feature sizes below 10 nm often require a roughness below 1 nm to ensure the reliability of the devices [16,17]. Only CMP can achieve such high precision in planarization [18,19,20]. This process is balanced to ensure that the surface is not excessively damaged by mechanical forces and that the chemical reaction proceeds uniformly across the surface. Following the implementation of an innovative slurry in the Si CMP process, the resultant surface roughness displayed an exceptional measurement of 0.067 nm [21]. Recently, we developed a dual-functional polishing slurry, incorporating ethylenediamine tetramethylphosphonic acid as a complexing agent and octahydroxamic acid as a highly effective corrosion inhibitor [22]. The application of this pioneering slurry in the context of Co interconnect CMP processes has yielded noteworthy outcomes, including a substantial removal rate of 246 nm/min and a root-mean-square roughness measuring 2.31 nm. Therefore, CMP could be considered one of the effective strategies for addressing nanoscale surface roughness in biomaterials at the nanometer scale.

In this study, we prepared an environmentally friendly polishing fluid, containing 4 wt% SiO_2_, 0.3 wt% H_2_O_2_, 1.0 wt% glycine, and 0.05 wt% BTA, for polishing the CoCrMo alloy used for artificial joints, achieving a very low arithmetic mean roughness of 0.25 nm at a relatively high removal rate. This was due to the efficient chelation of the glycine with the alloy and the inhibitory effect of the BTA. This method offers a straightforward, efficient, and highly cost-effective approach to reduce roughness, optimize friction behavior, and potentially extend functional longevity. Consequently, this research introduces a fresh perspective on and strategy for the planarization treatment of joint prostheses, with potential applications in other fields of medical material surfaces.

## 2. Materials and Methods

### 2.1. Materials 

A colloidal silica (mean particle size ∼60 nm, pH ∼9.4, Jin Wei Group Co., LTD, Ningbo, China) was selected as the abrasive in the slurry for the polishing experiment. The CoCrMo alloy, composed of a Co matrix, 28% Cr by mass, and 5% Mo by mass, was supplied by Tianjin Just Medical Technology Group Co., LTD, Tianjin, China. This specific composition of the CoCrMo alloy results in a density that typically approximates 8.94 g/cm^3^. 

### 2.2. Slurry Preparation

We chose colloidal silica as the abrasive component for the polishing experiment, ensuring that the remaining experimental solutions remained devoid of it. Our base solution comprised deionized water and 30% H_2_O_2_ (mass fraction). Subsequent pH adjustment to 8.0 was accomplished via the addition of KOH and diluted HNO_3_. Lastly, the complexing agent (glycine, supplied by Macklin) and the inhibitor (BTA, supplied by Macklin, Shanghai, China) were integrated into the polishing slurry.

### 2.3. Polishing Experiments

The polishing regimen was implemented via an E460 polishing machine (Alpsitec Inc., Vinoux, France) using a Politex polishing pad from Dow Chemical Company, Midland, MI, USA. The polishing substrate was a CoCrMo alloy disk with a radius of 3.8 cm. The specific parameters included a polishing pressure of 1.5 psi, a slurry flow rate of 0.3 L/min, a polishing head speed of 87 rpm, and a polishing disc plate speed of 93 rpm. To mitigate inter-group discrepancies, the polishing pad was pre-conditioned using a diamond dresser for 300 s before each polishing sequence. Post-polishing, the CoCrMo alloy disk was rinsed with deionized water and dried using high-purity N_2_ steam. A Mettler Toledo AB204-N analytical balance, with an accuracy of 0.1 mg, was used to measure the weight change pre- and post-polishing to compute the cobalt disc removal rate. Each recorded weight value was given as the average of three trials in order to minimize measurement error.

To quantify the removal rate during the CMP process, we employed the following equation:(1)$$\mathrm{removal}\;\mathrm{rate}=\frac{\Delta m}{\rho \times A\times t}$$

### 2.4. Electrochemical Measurements

We employed a CHI660E electrochemical workstation with a three-electrode cell setup to evaluate the electrochemical properties of the CoCrMo alloy disk electrode in solution. The system comprised a CoCrMo working electrode, a platinum counter electrode, and a saturated calomel electrode (SCE) as a reference. Prior to testing, the CoCrMo electrode was sequentially polished using 1500#, 2500#, and 3500# SiC sandpaper, then rinsed with deionized water and dried using high-purity N_2_ steam. The potential polarization curve parameters were as follows: E_ocp_ ± 0.3 V open circuit voltage, 5 mV/s voltage scan rate.

### 2.5. Characterization Methods

To remove the CoCrMo oxide film, each set of CoCrMo samples was immersed in a slurry solution having a concentration of 25 mM for 10 min, following which the freshly exposed Co surface was rinsed with deionized water and dried using high-purity N_2_ steam. Each sample underwent immersion in various solutions for 10 min, followed by repeated rinsing and drying. Subsequent analyses of surface morphology, arithmetical mean height, and elemental composition were performed using an AFM (5600LS, Agilent, Santa Clara, CA, USA), an SEM (Sigma 500, ZEISS, Baden-Württemberg, Germany) system, and an XPS (ESCALAB250Xi, Thermo, Waltham, MA, USA) system. The XRD patterns of the CoCrMo alloy were recorded on an X’Pert diffractometer equipped with graphite-monochromatized Cu K radiation.

### 2.6. Finite Element Analysis

Finite element analysis (FEA) was conducted, employing COMSOL Multiphysics 6.0 software, to simulate the mechanical behavior of knee joint motion using artificial implants. A three-dimensional model encompassing the geometries of standard tibia and femur bones, along with the prosthetic knee components, was established. The material properties were assigned to three distinct materials: bone, high-density polyethylene (HDPE), and CoCrMo alloy, with the bone’s properties sourced from the COMSOL Material Library.

The Solid Mechanics Interface was utilized to approximate the domains as linear elastic, reflecting the mechanical interactions with the introduction of artificial knee joints. Initial conditions were set to zero, indicating no pre-existing movement before bending. To streamline the simulation, the motion of the femur and femoral prosthesis was fixed, creating a reference frame, while allowing the tibia and the membranous pad to articulate. This configuration enabled the rotation axis base point, direction, and speed to emulate real-case scenarios. The rotational speed was set to 2 rad/s to represent the knee’s motion during vigorous walking activities, acknowledging that peak knee flexion angular velocity typically ranges from 1 rad/s during a normal walk to 2 rad/s for brisk walking. 

Contact mechanics were integral to the simulation, allowing for interaction between the joint’s components. The normal contact force, Tn, was computed based on the contact pressure, pn, and the interfacial gap, g, adhering to the condition Tn = if (g ≤ 0, −png, 0). This indicates force application only upon a zero or negative gap, and it was deliberately set to a marginally negative value to ensure contact. The contact pressure pn was moderated by the contact pressure penalty factor fp = 0.01, suggesting a comparatively soft interaction.

The Coulomb friction model was selected, with a uniform surface structure at the joint interface assumed for simplicity. The friction coefficient, μ, was defined as 0.4 to depict the frictional force’s dependency on the normal force. The cohesion sliding resistance Tcoh was established at 60,000 N/m^2^ to characterize the initial resistance against sliding motion due to cohesive forces. Upon specifying boundary conditions and initial settings, the model was discretized using a mesh and subjected to a stationary study. The primary evaluation metric was the frictional force, which was extracted and scrutinized post-simulation for further insights into the joint’s performance under simulated conditions.

## 3. Results and Discussion

### 3.1. Surface Morphology of an Artificial Knee Joint

Figure 1a presents an artificial joint, composed of the femur, tibia, and patella, engineered to mimic the fluid motion of natural joints. The femur main body is constructed from a CoCrMo alloy, while durable hard HDPE substitutes for cartilage on the upper surfaces of the tibia and patella. Notably, one of the main causes for the failure of these artificial joints is the interface wear between the alloy and the HDPE.

Figure 1b,c provide SEM images of the HDPE pad on the surface of a failed tibial component from an artificial joint. Distinctly visible in Figure 1b are micrometer-sized particles situated around the scratches, with enough to precipitate joint failure. These particles are a consequence of the joint friction that causes wear at the joint interface. This is primarily adhesive and abrasive wear, and this wear typically results in a proliferation of these wear particles over time [23,24]. The presence of these particles can induce cell death around the joint prosthesis and even cause resorption of the bone surrounding the prosthesis. Moreover, high-frequency frictional movement between these particles and the alloy–HDPE interface engenders severe scratches on the HDPE pad, as depicted in Figure 1c. This scratching significantly contributes to inflammation within the joint area, thereby constituting another pivotal factor leading to artificial joint failure.

### 3.2. Chemical Mechanical Polishing of CoCrMo Alloys

To counteract this problem, the CMP method was employed. Following slurry composition optimization based on our previous research in the field of CMP [22,25], we utilized a solution consisting of 4 wt% SiO_2_, 0.3 wt% H_2_O_2_, 1.0 wt% glycine, and 0.05% BTA, where the SiO_2_ served as the abrasive, the H_2_O_2_ functioned as the oxidizer (Figure 2a,b), the glycine acted as a chelating agent, and the BTA was employed as an inhibitor. The interplay and specific mechanisms by which the oxidizer, chelating agent, and inhibitor contribute to the CMP process are elucidated in the subsequent section.

As observed in Figure 2c, and further detailed in its magnified counterpart (Figure 2d), the CoCrMo alloy surface post-CMP is smooth and devoid of scratches. Importantly, a smooth surface is advantageous in terms of reducing wear and tear, minimizing particle generation, improving biocompatibility, and enhancing mobility and comfort. Therefore, improving the surface properties of artificial joints can play an instrumental role in enhancing their functionality, their durability, and the overall success rate of joint replacement surgeries.

### 3.3. Investigating the Impact of Slurry Constituents on the Polishing Process 

The glycine and BTA were incorporated as complexing agents and inhibitors, respectively, in the polishing slurry, as depicted by their chemical structures in Figure 3a,b. The interaction dynamics between the various components of the polishing slurry and the CoCrMo alloy were explored through a removal rate study. As illustrated in Figure 3c, a relatively low removal rate of the CoCrMo alloy was achieved with the presence of the SiO_2_ abrasive alone. However, a significantly higher removal rate is requisite for the commercialization of CoCrMo prostheses. Even though solely mechanical polishing yielded a Ra of 0.51 nm (Figure 4a), the incorporation of H_2_O_2_ into the polishing solution led to a decrease in the CoCrMo alloy removal rate to 8.6 nm/min. This was attributed to the oxidation effect induced by the H_2_O_2_, forming an oxide film on the alloy surface that results in alloy passivation and localized excessive oxidation, thereby increasing the surface roughness [26].

The introduction of glycine substantially amplified the CoCrMo alloy removal rate. In its role as a complexing agent in CMP, interactions are formed between the functional groups of glycine (carboxyl and amino groups) and the metal ions in the alloy [27,28]. Initially, coordination bonds are created between the carboxyl and amino groups of glycine and the Co ions, with the metal ions serving as Lewis acids and bonding with the lone electron pairs on glycine’s functional groups. This coordination bonding gives rise to stable metal–glycine complexes, which subsequently yield chelating rings with the metal ions at their center, surrounded by glycine molecules. These resulting metal complexes are more stable than the original metal ions, inhibiting their re-dissolution or re-deposition during polishing. However, the robust chelation effect of the complexing agent escalated the arithmetic mean roughness (Ra) to 0.25 nm. 

To uphold the CoCrMo alloy surface quality in this CMP process, BTA is added, proving critical for enhancing the longevity of artificial knee joints. The BTA functions as an effective corrosion inhibitor, hindering undesirable surface reactions [29,30]. When the alloy is immersed in a BTA-containing solution, a passive layer is formed, comprising a complex between the alloy and the benzotriazole. This passive layer, insoluble in aqueous and numerous organic solutions, possesses a thickness that is positively correlated with the prevention of corrosion. Although the BTA slightly attenuated the polishing rate from 43.1 nm/min to 30.9 nm/min, it substantially improved the surface quality of the CoCrMo alloy, mitigating the Ra from 20.76 nm to a mere 0.25 nm (Figure 4d). This ultra-low surface roughness contributes to smoother performance and extended lifespan for artificial knee joints, effectively curbing the particle abrasion and mitigating the particle-induced inflammation and bone dissolution. 

### 3.4. Investigating the Average Particle Size of Colloidal Silica

The SiO_2_ abrasive we utilized is depicted in Figure 5a. The SEM revealed that the SiO_2_ nanoparticles were globular in shape, with an average particle size of approximately 60 nm, and they exhibited monodispersity. Aiming to delve further into the SiO_2_ collosol and the interplay between its chemical components, measurements of the mean particle size of the slurries were undertaken, as depicted in Figure 5b. The mean particle sizes of the four slurries, marked at 58.3, 57.2, 59.2, and 58.0 nm, respectively, display negligible variation. This suggests that the inclusion of H_2_O_2_, glycine, and BTA does not induce the aggregation of the SiO_2_ abrasive particles. The aggregation of SiO_2_ particles can induce surface scratching on the CoCrMo alloy, so the observed decrease in surface roughness brought about by H_2_O_2_ and glycine is attributed to their corrosive and oxidative effects.

### 3.5. Investigating the Chemical States and Crystalline Structure of the CoCrMo Alloy

The influence of the H_2_O_2_, glycine, and BTA on the phase structure of the CoCrMo alloy was scrutinized using X-ray diffraction (XRD) analysis. Figure 6a indicates that the XRD patterns of the CoCrMo alloy persisted unaltered post-treatment with diverse slurries. This hints that neither the oxidative action of the H_2_O_2_, the complexing function of the glycine, nor the inhibitory role of the BTA modifies the phase structure of the CoCrMo alloy itself; these chemical reactions are restricted to the surface. The XRD pattern manifests peaks associated with metallic Co, with the peak at 44.2° linked to the (111) crystal plane of Co, in alignment with the PDF card PDF#15-0806 [31]. Such an occurrence is due to the formation of a solid solution by the metals in the alloy, which are fully soluble in each other. Therefore, the XRD pattern primarily mirrors the crystal structure of the solvent metal, particularly when the quantity of the solute metal is minor. The homogeneous distribution of the Mo and Cr elements in the CoCrMo alloy was verified using element mapping, as presented in Figure 6b. Despite their lower content compared to Co, the Mo and Cr elements were uniformly dispersed in the CoCrMo alloy. Consequently, the concentration of the MoCr metal may have fallen beneath the detection limit of the employed XRD technique, resulting in the exclusive manifestation of the predominant Co metal in the XRD analysis.

Surface characterization of the CoCrMo alloy, within a depth range of approximately 10 nm, was performed using X-ray photoelectron spectroscopy (XPS). Figure 7a exhibits the Co 2p XPS spectra, where the peak at a binding energy of 778.3 eV corresponds to Co^0^. When only SiO_2_ abrasive populates the polishing solution, the polishing process engages merely mechanical forces, keeping the valence state of Co in the CoCrMo alloy unaltered, in congruence with the XRD results. However, with the introduction of H_2_O_2_ into the polishing solution, oxidation of the Co commences, initiating the formation of Co_3_O_4_ and Co(OH)_2_, as denoted by peaks at 779.7 eV and 781.6 eV, respectively [32,33]. Conversely, with the inclusion of glycine, no peaks associated with Co(OH)_2_ are discernible in the Co 2p_3/2_ XPS spectra, indicating an effective glycine complexation with the Co(OH)_2_. Upon further addition of BTA to the slurry, peaks corresponding to Co(OH)_2_ reemerge in the XPS spectra, implying that the CoCrMo alloy surface can effectively adsorb the BTA, thus curtailing the complexation effect of the glycine. Additionally, Figure 7b illustrates the O 1s peaks at 530.2 eV and 530.7 eV, corresponding to Co_3_O_4_ and Co(OH)_2_, respectively. These findings are consistent with the previously discussed Co 2p_3/2_ results.

### 3.6. Investigating the Corrosion and Corrosion Inhibition Mechanisms of the CoCrMo Alloy

Electrochemical corrosion tests, carried out using a three-electrode setup, as demonstrated in Figure 8a, were employed to probe the complexation effect of the glycine and the inhibitory effect of the BTA. Figure 8b illustrates that the introduction of the glycine escalated the corrosion current density (I_corr_) to 9.55 µA/cm^2^, reduced the corrosion potential (E_corr_) to 0.16 V, and shifted the potentiodynamic polarization curve towards the bottom right. These alterations are credited to the glycine complexing with the ionized Co on the oxidized oxide layer of the CoCrMo alloy, precipitating the formation of Co complexes, and thus promoting surface erosion of the CoCrMo alloy. With the subsequent introduction of the corrosion inhibitor BTA, both E_corr_ and I_corr_ of the potentiodynamic polarization curve decreased to 0.18 V and 4.49 µA/cm^2^, respectively. This transition is ascribed to the effective inhibition of the cathode reaction by the BTA, leading to the displacement of the cathode branch towards a lower corrosion current density.

### 3.7. The Mechanisms of Chemical Mechanical Polishing in CoCrMo Alloys

The enhancement in surface quality and reduction in roughness of CoCrMo alloys, while ensuring adequate removal rates, is achieved through four principal mechanisms in the CMP process, which are predominantly simultaneous (Figure 9). Initially, the H_2_O_2_ acts as an oxidizing agent, instigating an oxidation reaction with the Co, the principal component of the CoCrMo alloy, yielding Co_3_O_4_ and Co(OH)_2_. Subsequently, the glycine forms complexation reactions, yielding [Co(glycine)^3^] and [Co(glycine)^4^] complexes, respectively. Concurrently, the BTA physically and chemically adheres to the surface of the CoCrMo alloy, curtailing the corrosive action of the glycine, thereby constraining the rise in surface roughness and inhibiting the emergence of defects. Lastly, the mechanical exertion of the SiO_2_ abrasive eradicates the surface oxide layer, delivering a surface with a Ra of 0.25 nm.

### 3.8. Simulation of Frictional Forces

To investigate the influence of the superior surface quality of CoCrMo alloy on the lifespan of artificial joints, the frictional forces at the interface between CoCrMo and HDPE were computed using a Multiphysics simulation (Figure 10a) [34]. The results, depicted in Figure 10b, demonstrate a significant decrease in these forces. This reduction is a direct consequence of the improved surface quality achieved through chemical mechanical polishing, resulting in nano-smooth CoCrMo alloy surfaces.

As a result of the decreased frictional forces, there is a more uniform distribution of pressure across the pad, particularly in the contact region between the tibial pad and the femoral prosthesis. This uniformity indicates that optimal alignment conditions are essential to minimize wear in high-stress regions, especially in the contact zone. Our graphical analysis further supports this finding, showing a decrease in the frictional forces from initial values of 48.3, 48.1, 29.3, and 43.9 N/cm^2^ to 17.6, 17.5, 12.2, and 18.5 N/cm^2^. These data suggest that smoother surfaces with lower roughness significantly reduce wear in stress-concentrated areas of the prosthesis pad, thereby diminishing stress shielding after knee arthroplasty.

According to Wolff’s Law, the higher the strength of the joint prosthesis material and the more inappropriate the placement of the prosthesis, leading to an uneven distribution of equivalent stress, the more the load on the osteotomy surface behind the prosthesis decreases [35]. This uneven distribution, manifesting as extremely high or low stress, known as stress shielding, can influence the remodeling of the corresponding cancellous bone. Insufficient or excessive forces stimulating bone growth can lead to bone remodeling in the respective areas, altering the original bone density and causing a shift in the force transmission of the knee joint. These factors interact, resulting in osteoporosis and bone atrophy in non-stress-concentrated areas, leading to loosening or even dislocation of the knee joint prosthesis, necessitating revisions and ultimately reducing the prosthesis’s operational lifespan.

## 4. Conclusions

In conclusion, our research presents an effective CMP slurry formulation, comprising 4 wt% SiO_2_, 0.3 wt% H_2_O_2_, 1.0 wt% glycine, and 0.05 wt% BTA, specifically designed for refining CoCrMo alloy surfaces. This formulation achieves a notable material removal rate of 30.9 nm/min and attains an arithmetic mean surface roughness of just 0.25 nm, demonstrating the synergy between the chemical and mechanical aspects of the CMP process. The exceptional removal rate is largely attributed to the potent complexation activity of the glycine, while the significantly reduced surface roughness results from the inhibitory action of the BTA. The XPS analyses revealed surface chemistry dynamics, emphasizing oxidation, complexation processes, and the BTA’s protective role against corrosion in a single, comprehensive narrative. The preservation of the alloy’s phase structure, confirmed by the XRD analysis and elemental mapping, ensures the mechanical integrity of the treated surfaces. The electrochemical tests further reinforced this, showing a decrease in corrosion potential (E_corr_) to 0.18 V and in corrosion current density (I_corr_) to 4.49 µA/cm^2^ with the addition of BTA, achieving a nano-smooth CoCrMo alloy surface. These outcomes align with our computational findings and highlight the efficacy of our CMP method in improving artificial knee joint surfaces, potentially extending their lifespan and reducing complications. This study significantly advances biomaterial science, offering a novel surface-treatment strategy that blends chemical innovation with mechanical precision.

## Figures and Tables

**Figure 1 materials-17-00008-f001:**
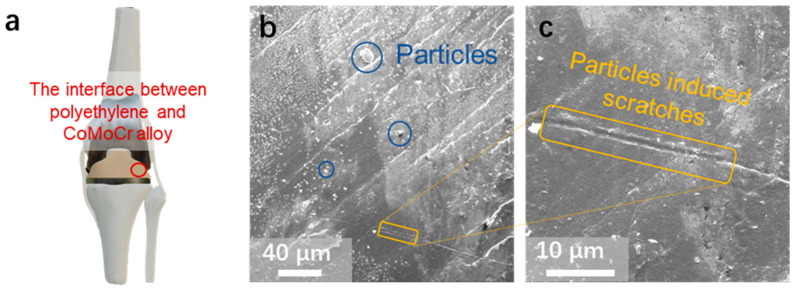
(**a**) Schematic representation of an artificial joint. (**b**) SEM image of a worn PE pad extracted from a used artificial joint. (**c**) Magnified view of the wear area from (**b**).

**Figure 2 materials-17-00008-f002:**
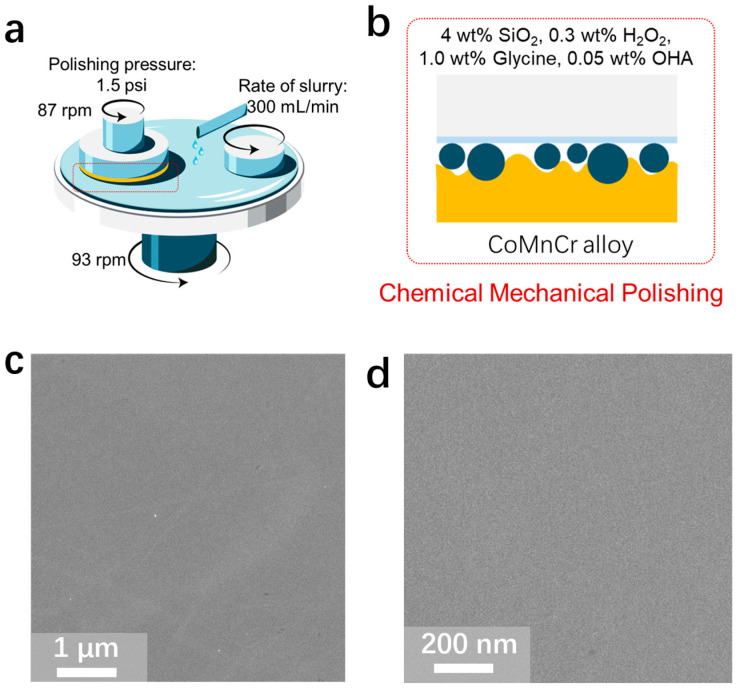
(**a**) Schematic diagram illustrating the CMP process. (**b**) Conceptual illustration of the interface composition of the polishing fluid as depicted in (**a**). (**c**) SEM image of the polished CoCrMo alloy. (**d**) Magnified view of the wear area from (**c**).

**Figure 3 materials-17-00008-f003:**
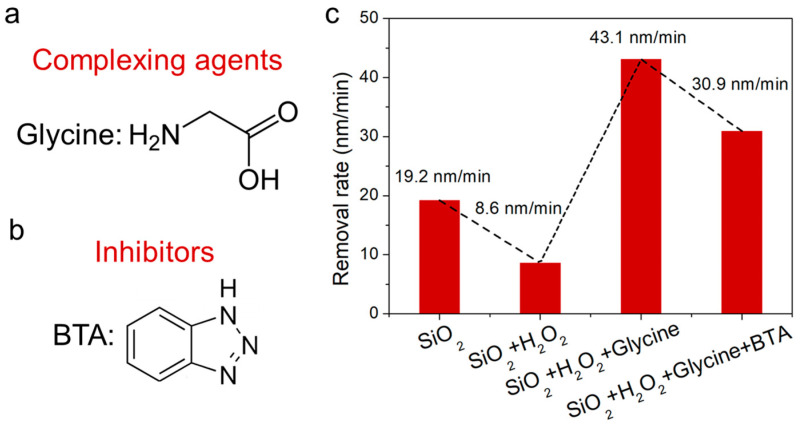
(**a**) The molecular structure of glycine and (**b**) BTA. (**c**) An examination of the removal rate for the slurry compositions including SiO_2_, SiO_2_ + H_2_O_2_, SiO_2_ + H_2_O_2_ + glycine, and SiO_2_ + H_2_O_2_ + glycine + BTA.

**Figure 4 materials-17-00008-f004:**
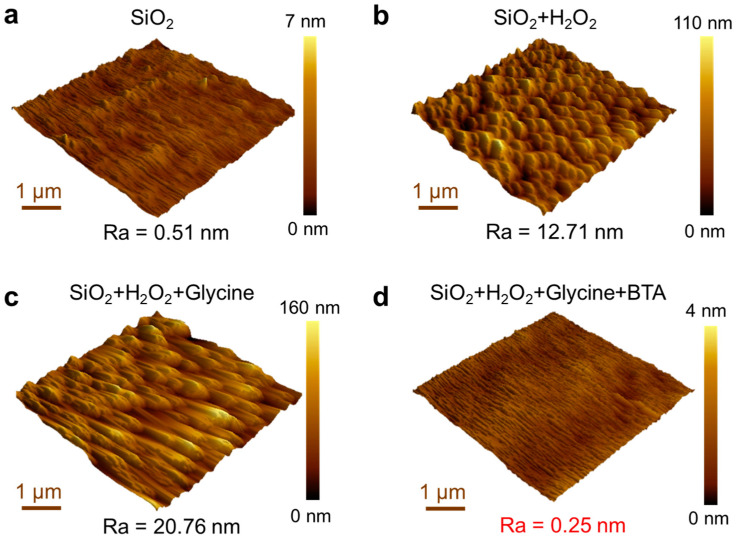
The arithmetic mean roughness for the slurry compositions including (**a**) SiO_2_, (**b**) SiO_2_ + H_2_O_2_, (**c**) SiO_2_ + H_2_O_2_ + glycine, and (**d**) SiO_2_ + H_2_O_2_ + glycine + BTA.

**Figure 5 materials-17-00008-f005:**
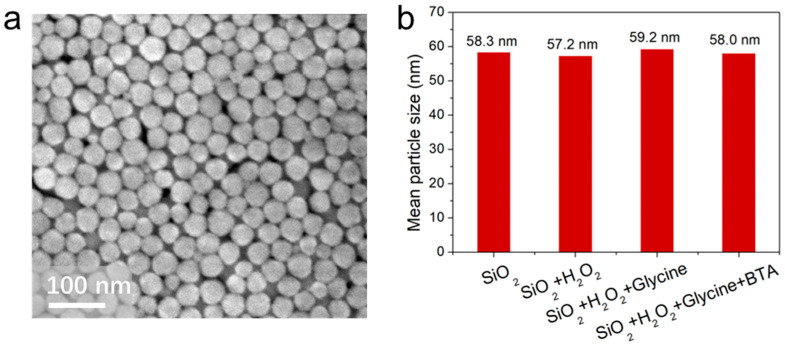
(**a**) SEM image of SiO_2_ abrasive. (**b**) The average particle size of SiO_2_, SiO_2_ + H_2_O_2_, SiO_2_ + H_2_O_2_ + glycine, and SiO_2_ + H_2_O_2_ + glycine + BTA.

**Figure 6 materials-17-00008-f006:**
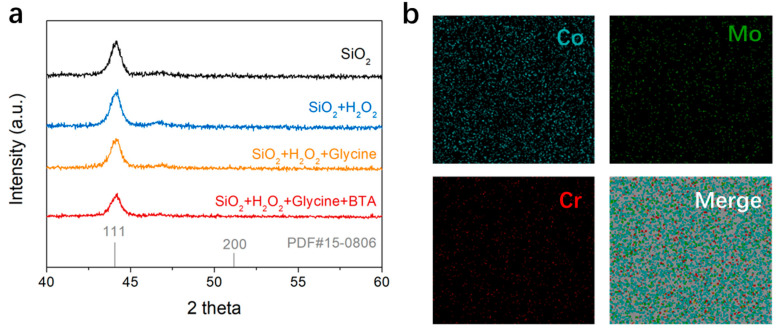
(**a**) XRD pattern of slurries containing SiO_2_, SiO_2_ + H_2_O_2_, SiO_2_ + H_2_O_2_ + glycine, and SiO_2_ + H_2_O_2_ + glycine + BTA. (**b**) An elemental mapping of the CoCrMo alloy after CMP.

**Figure 7 materials-17-00008-f007:**
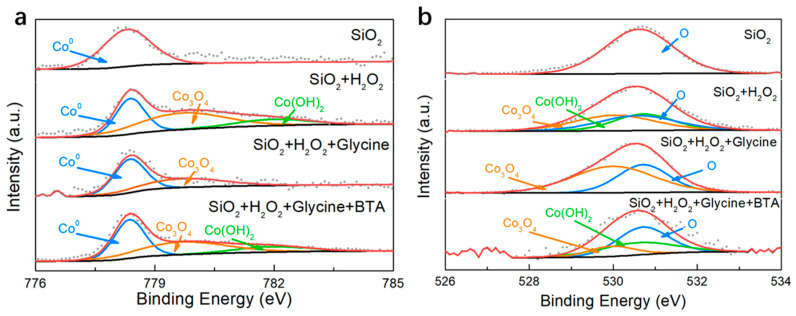
(**a**) Co 2p and (**b**) O 1s XPS spectra of CoCrMo alloy.

**Figure 8 materials-17-00008-f008:**
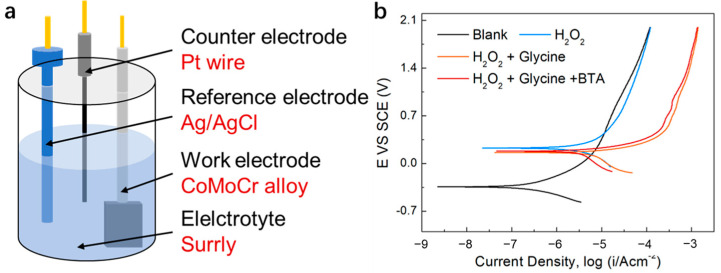
(**a**) A schematic representation of the electrochemical corrosion test. (**b**) Potentiodynamic polarization curves for the CoCrMo alloy in different slurry conditions.

**Figure 9 materials-17-00008-f009:**
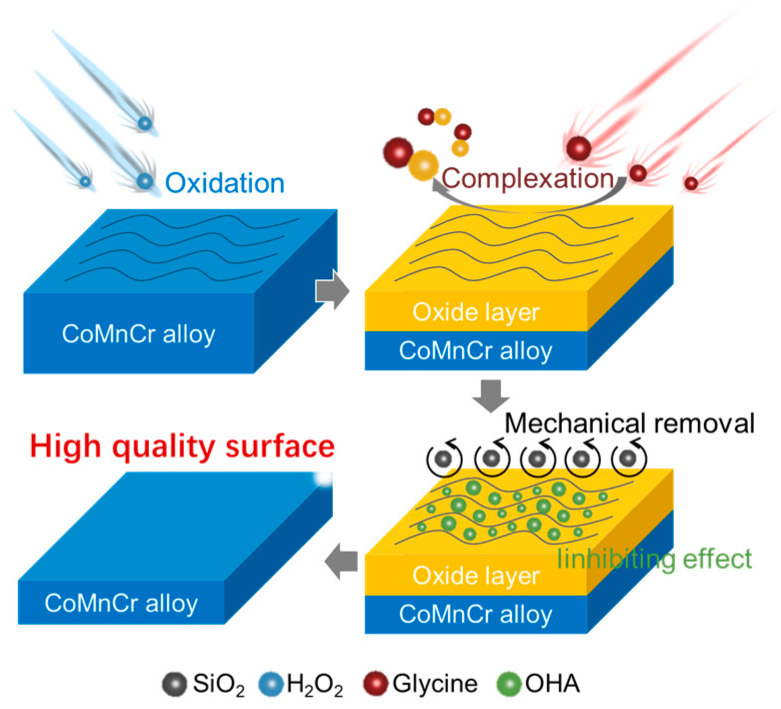
A schematic visualization of the CMP process for the CoCrMo alloy.

**Figure 10 materials-17-00008-f010:**
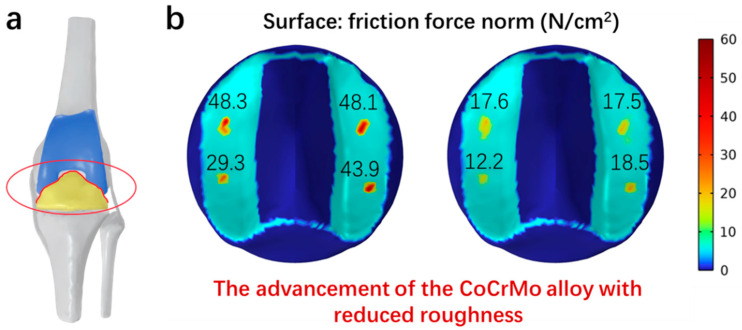
(**a**) Model of the artificial knee joint interface. (**b**) Simulation of frictional forces between CoCrMo alloy and HDPE at the red interface shown in (**a**) marked by the red circle.

## Data Availability

Data are contained within the article.

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
