# Peer review of "Nanoscale Surface Refinement of CoCrMo Alloy for Artificial Knee Joints via Chemical Mechanical Polishing"

_materials, 2023, doi:10.3390/ma17010008_

Round 1
Reviewer 1 Report
Comments and Suggestions for Authors
Revision of the manuscript entitled “Chemical Mechanical Polishing of CoCrMo Alloy: Beneficial for the Friction Behavior of Artificial Knee Joints” (2731567).
Comments:
Although the study does not present deep investigations about the genesis of the polishing process, the results support the hypotheses presented. The subject is very pertinent and interesting for the tribology field of biomaterials. However, the manuscript needs minor and major revisions to be accepted for publication.
Major revision
· To discuss topography changes of materials it is crucial to define the initial condition of the samples. In the manuscript there is no information about the initial samples' topography, samples' chemical composition, or even how the samples were produced, acquired, who is the manufacturer.
· The results from XRD analyses were reported in the results, but it is not presented in the materials and methods section.
· In section 3.1, the information contained between line 116 to 136 are literature information, therefore it can be replaced to the introduction section. Additionally, the content in the last paragraph of this section repeats information already mentioned in the materials and methods section.
· The bar graphs do not present the dispersion of the results.
· It is not clear how the material removal was calculated, especially because the CoCrMo alloy density is unknown.
· Why did the authors use the Sq parameter to characterize the topography, how it was calculated? I suggest using more than one parameter like Sa, Sk, Sz.
· In line 165, the authors mentioned the formation of a “dense oxide layer”, but normally the metal oxide density is lower than metals. In addition, this layer can present a certain porosity.
Minor revision
· The figures contain too much information in the same figure. I suggest separating them in more in order to clarify the information of each figure. Additionally, the numbers and letters in the figures are too small compared to text letters.
· Could you present more details about the model used to study the contact conditions? Does the model consider the formation of debris?
The study is interesting, but it needs to improve significantly. Therefore, I recommend accepting the manuscript for publication with major revision.

Reviewer 2 Report
Comments and Suggestions for Authors
In materials-2731567 submission, the friction behavior of artificial knee joints after the chemical mechanical polishing of CoCrMo alloy is described. This subject is actual and interesting, however, some remarks should be done:
1. Fig. 1 a and 1 d should be transferred to section 2 and described there,
2. Surface roughness is a concept and has no units of measurement
3. Sq is untypical roughness parameter; why the Authors didn’t study Ra, Rz etc. Surface texture parameters (S) have nothing to do with roughness (R),
4. Multiphysics simulation idea and results should be described in detail,
5. Conclusions should be extended,
6. Typos should be corrected.
Reviewer 3 Report
Comments and Suggestions for Authors
This manuscript cannot be accepted in its present form and needs extensive revision. My comments are given below.
1. The details and comparative discussion about CMP process with other methods are not included in the introduction part. Also, add some literature done by this process and discuss the expected outcome of this process.
2. The choice and ratio of selecting SiO2, H2O2, glycine are not clear.
3. Is it possible to polish the proposed artificial joint made of CoMoCr alloy proposed in this study using this CMP process? Because the author chose circular disc with a flat bottom, but the actual component is not the flat one. Justify your choice in selecting this polishing process.
4. Provide high magnification image using HRSEM instrument. Because the authors claim the surface roughness of 0.3 nm. It is better to see the microstructure in a highly magnified image.
5. Please confirm whether the high magnification image (Figure 1c) was taken from the rectangle as shown in Figure 1b. Because in Fig. 1b, the scratch projected downside and in Fig.1c it is projected slightly upside.
6. Line 137-144 in the result and discussion section (Page 4) is more suitable to be in the introduction section.
7. Include the roughness profile images from the AFM data. 0.3 roughness value is lesser than 0.5 nm roughness value of commercially available well-polished silicon wafer.
8. Why is the XRD pattern of alloy (111) peak too broad, and no peak corresponds to (200) appears? Is it a single crystal alloy?
9. XPS peak fitting (Figure 3e) seems very broad. What is the FWHM value of Co, Co3O4 and Co2O3? Where are 2p3/2 and 2p1/2 peaks? 780.4 eV may be due to CoOOH formation. Please refer for fitting and interpretation 10.1039/C9DT02301A
10. Provide XPS survey spectra and also Oxygen high resolution spectra.
11. While measuring PDP analysis please measure up to breakdown potential. Then only we can understand the oxide layer formation and stability of the alloys under corrosion.
12. Overall, the experimental section is weak. If the author proposes this work for tribological application, the author should have provided friction coefficient data. Without this data, the work is incomplete.
Reviewer 4 Report
Comments and Suggestions for Authors
Dear Authors,
The paper contributes to the advancement of the field of tribological properties of CoCrMo alloy, a crucial area of study for the development of improved implants in which tribological contact is unavoidable. However, the relevance of the work and the experimental and characterization efforts that were conducted are not adequately emphasised in the paper's format. For example, the authors have employed a range of characterization techniques (e.g., XPS, XRD, SEM, corrosion studies, simulations, etc.) in the current investigation. However, they have not provided comprehensive explanations regarding the rationale behind their selection of these methodologies nor the potential and noteworthy findings that may be derived from their combined efforts. Even the abstract does not adequately represent the content of the work. The introduction merely provides an outline of the specific topic and fails to emphasise the significance of the current investigation or the current state of the art. Additionally, the results and discussion sections must be revised to include more precisely express the study's findings. In conclusion, I regret to notify you that the paper does not meet the necessary standards to be considered for publication. The Authors have to make substantial improvements and if possible, resubmit it following the reviewers comments and suggestions. In addition to these overall remarks, I would want to provide you with section-specific suggestions and input for each specific part of the article.
1. Title should be more specific to address the novelty of the paper. Please revise the title.
2. The abstract stimulates interest, but it needs to present some more significant findings of the study, such as variation of wear mechanisms as a function of chemical polishing, XPS results, particle sizes, simulation results, etc.. The authors should present some key finding, if possible provide some more quantitative results. In the current form, the abstract includes vague findings. Also, the scientific and engineering impact of the study should be emphasized.
3. The punctuation should be used after the citations. For example Artificial knee joint failure can be attributed to numerous causes, among which severe joint wear stands out as a major contributor [1-3].
4. The introduction describes the content in a relationship with the aim of the study. It builds interest in the topic. However, the relevant papers similar to this study need to be cited and briefly discussed. Besides, the introduction includes much general and well-known information about the topic, which can be omitted or shortened. To sum up, a brief section discussing the recent studies similar to the study needs to be included.
5. Materials and methods section needs to be improved to provide enough details that another researcher could replicate the process. This section should include subsections, for instance tribological characterisation, in which the details of each characterisation methods should be clarified and detailed.
6. The experimental procedures and details should be given in its respective section. For instance, the details of chemical polishing given in Figure 1-d and -e need to be given in section 2.
7. Please enlarge SEM images in Figure 1 so that the readers can see the morphology of the surfaces after polishing.
8. In Figure 1-a, Schematic representation of an artificial joint should be cited if its not belong to the authors original drawings.
9. Figure 2 should be separated into 2 figures. Please provide the surface topographies with larger sized in a separate image.
10. Again Figure 3 should be separated into 3 figures. It is not possible to read the details. Besides, it is not insightful to present SEM, particle size analysis, XRD, EDS, and corrosion data in the figure. Please prepare new Figures.
11. Figure 5 has not been cited in the text. Please see section 3.4., please first cite the figure.
12. The authors give sufficient figures and visual material to discuss the topic as the visual content is helpful and worthy for publication. However, the results should be compared and discuss with the relevant studies already presented in the literature. Besides, the underlying reasons for the results should be emphasized by citing some fundamental studies. Please also follow my suggestions below:
13. It briefly recaps the main ideas by moving from specific to general ideas. The conclusion should better identify the overall conclusions and importance of the study for academic and industrial work. Also, a brief description of future work that can be followed after this study needs to be added.
The English needs minor corrections.
Round 2
Reviewer 3 Report
Comments and Suggestions for Authors
Thank you for your revision. I have a small concern. Please take a look at the XPS data fitting (Figure 7b). A entire region of one peak within another is not the correct method. Furthermore, the fitted peaks with FWHM greater than 3.5 are incorrect. Please re-fit the data and rewrite your XPS part. In addition, I was unable to locate the survey spectrum mentioned in the author's response.
Author Response
Thank you for your suggestion. We have revised the XPS section as per your recommendations. Please review the updated manuscript for the changes.
Reviewer 4 Report
Comments and Suggestions for Authors
Dear Authors,
I appreciate your attention to my suggestions and your completion of the necessary modifications. The paper has been significantly improved after these revisions.
Best wishes,
Reviewer
Comments on the Quality of English LanguageThe quality of the English language is adequate.
Author Response
We have made the necessary revisions to the English language in our manuscript. Please review the updated version for these changes.